# Estimation of Photosynthetic and Non-Photosynthetic Vegetation Coverage in the Lower Reaches of Tarim River Based on Sentinel-2A Data

Zengkun Guo [1,2] , Alishir Kurban [1,2,3,4,*] , Abdimijit Ablekim [1,2], Shupu Wu [1,2], Tim Van de Voorde [5,6] , Hossein Azadi [1,5,7] , Philippe De Maeyer [1,5,6] and Edovia Dufatanye Umwali [1,2]

1    Xinjiang Institute of Ecology and Geography, Chinese Academy of Sciences, Urumqi 830011, China; guozengkun18@mails.ucas.ac.cn (Z.G.); abdimijit@ms.xjb.ac.cn (A.A.); wushupu18@mails.ucas.ac.cn (S.W.); hossein.azadi@ugent.be (H.A.); Philippe.DeMaeyer@ugent.be (P.D.M.); umwariedovia@outlook.com (E.D.U.)
2    University of Chinese Academy of Sciences, Beijing 100049, China
3    Research Center for Ecology and Environment of Central Asia, Chinese Academy of Sciences, Urumqi 830011, China
4    Sino-Belgian Joint Laboratory for Geo-Information, Xinjiang Institute of Ecology and Geography, Urumqi 830011, China
5    Department of Geography, Ghent University, 9000 Ghent, Belgium; Tim.VandeVoorde@UGent.be
6    Sino-Belgian Joint Laboratory for Geo-Information, Ghent University, 9000 Ghent, Belgium
7    Faculty of Environmental Sciences, Czech University of Life Sciences Prague, 16500 Prague, Czech Republic
*    Correspondence: alishir@ms.xjb.ac.cn

**Abstract:** Estimating the fractional coverage of the photosynthetic vegetation ($f_{PV}$) and non-photosynthetic vegetation ($f_{NPV}$) is essential for assessing the growth conditions of vegetation growth in arid areas and for monitoring environmental changes and desertification. The aim of this study was to estimate the $f_{PV}$, $f_{NPV}$ and the fractional coverage of the bare soil ($f_{BS}$) in the lower reaches of Tarim River quantitatively. The study acquired field data during September 2020 for obtaining the $f_{PV}$, $f_{NPV}$ and $f_{BS}$. Firstly, six photosynthetic vegetation indices (PVIs) and six non-photosynthetic vegetation indices (NPVIs) were calculated from Sentinel-2A image data. The PVIs include normalized difference vegetation index (NDVI), ratio vegetation index (RVI), soil adjusted vegetation index (SAVI), modified soil adjusted vegetation index (MSAVI), reduced simple ratio index (RSR) and global environment monitoring index (GEMI). Meanwhile, normalized difference index (NDI), normalized difference tillage index (NDTI), normalized difference senescent vegetation index (NDSVI), soil tillage index (STI), shortwave infrared ratio (SWIR32) and dead fuel index (DFI) constitutes the NPVIs. We then established linear regression model of different PVIs and $f_{PV}$, and NPVIs and $f_{NPV}$, respectively. Finally, we applied the GEMI-DFI model to analyze the spatial and seasonal variation of $f_{PV}$ and $f_{NPV}$ in the study area in 2020. The results showed that the GEMI and $f_{PV}$ revealed the best correlation coefficient ($R^2$) of 0.59, while DFI and $f_{NPV}$ had the best correlation of $R^2 = 0.45$. The accuracy of $f_{PV}$, $f_{NPV}$ and $f_{BS}$ based on the determined PVIs and NPVIs as calculated by GEMI-DFI model are 0.69, 0.58 and 0.43, respectively. The $f_{PV}$ and $f_{NPV}$ are consistent with the vegetation phonological development characteristics in the study area. The study concluded that the application of the GEMI-DFI model in the $f_{PV}$ and $f_{NPV}$ estimation was sufficiently significant for monitoring the spatial and seasonal variation of vegetation and its ecological functions in arid areas.

**Keywords:** photosynthetic vegetation; non-photosynthetic vegetation; GEMI-DFI model; seasonal variation

## 1. Introduction

Vegetation cover is an important part of the ecosystem, which plays an important role in the soil and water conservation, restraining the desertification process, biodiversity protection and other ecological service and functions etc. [1,2]. From a functional point

of view, vegetation can be divided into two parts: the photosynthetic vegetation (PV) and the non-photosynthetic vegetation (NPV) [3]. The PV mainly refers green leaves, while the NPV mainly includes the vegetation litter (dead leaves, dead branches and stems) and crop residues [4,5]. NPV is important as it not only affects the carbon storage, $CO_2$ exchange capacity and temperature between the surface and soil [6,7] but also slows down soil erosion, increases the soil organic matter and improves the soil quality [8–12]. Therefore, an accurate estimation of the fractional coverage of the photosynthetic ($f_{PV}$) and non-photosynthetic vegetation ($f_{NPV}$) is of great significance in evaluating the ecological conditions.

In recent decades, remote sensing technology has made great progress in many aspects of geography [13,14]. In the $f_{PV}$ estimation, a number of indices (e.g. normalized difference vegetation index (NDVI) [15], soil adjusted vegetation index (SAVI) [16] and global environment monitoring index (GEMI) [17]) have been developed for PV utilizing spectral reflectance, which is widely used in various satellite platforms. Researchers have also proposed some vegetation indices so as to estimate the $f_{NPV}$ [18–23]. According to the spectral resolution of remote sensing data, non-photosynthetic vegetation indices (NPVIs) can be divided into hyperspectral NPVIs and multispectral NPVIs. Daughty [21] proposed hyperspectral NPVI (cellulose absorption index (CAI)) based on hyperspectral reflection characteristics of the NPV. Taking into account the limitations of hyperspectral data, researchers have developed many multispectral NPVIs using Landsat TM bands, such as the normalized difference index (NDI), the normalized difference tillage index (NDTI), the normalized difference senescent vegetation index (NDSVI) and the soil tillage index (STI). Guerschman [3] developed the shortwave infrared ratio index (SWIR32) based on MODIS data, which accurately estimated the $f_{NPV}$ of grasslands in Australia. Cao [24] utilized multispectral, MODIS, data to estimate NPV using the dead fuel index (DFI).

In order to estimate the proportion of PV and NPV in a pixel, based on spectral mixture analysis (SMA), researchers construct pixel unmixed model by using PVI and NPVI to estimate $f_{PV}$ and $f_{NPV}$. And based on this, Guerschman [3] used the Hyperion data to propose pixel linear unmixed model of the NDVI-CAI (NDVI represents the PV, CAI represents the NPV). This updated methodology includes a better estimation of the spatiotemporal distribution of the PV and NPV in the sparse grasslands of Australia. Validating this methodology, Wang [25] applied the DFI index to construct the NDVI-DFI linear unmixed model based on the MOD09GHK surface reflectance data and effectively calculated the $f_{PV}$ and $f_{NPV}$ of the Xilingol grasslands. The foundational work mentioned above successfully showed suitable PV and NPV estimation using multispectral data but were limited in geographic and ecological scope. Focusing primarily on grasslands, other regions have not been explored yet. This work aims to estimate $f_{PV}$ and $f_{NPV}$ in riparian regions.

The Tarim River is a typical arid inland river basin. Due to the impact of human reclamation and irrigation in the upper reaches of the Tarim River, the water of the lower reaches has dropped sharply over the past 20 years, the Tetima Lake has dried up and the vegetation environment has deteriorated year by year. A comprehensive assessment of the vegetation changes in this area has been an ongoing effort. Some of the work, by Guli [26] and Li [27] employed a variety of models to discuss the suitability of the estimated vegetation coverage in the Tarim River Basin. Zhu [28] applied the method of transfer learning so as to extract and analyze the vegetation coverage changes in a long time series. These studies neglect to include NPV data in their analyses and rely largely on course resolution imagery, resulting in limited accuracy and uncertainty in the original estimations. Considering the improvements made to spectral instruments and the requirements for time and spatial resolution, Sentinel-2 satellite data are a good choice, specifically the multispectral imager (MSI) onboard could provide multispectral data with spatial resolution of 10 m and revisiting period of five days. Therefore, using the Sentinel-2 data with a high spatial and temporal resolution will be the focus to estimate the $f_{PV}$ and $f_{NPV}$ in the study area.

The objective of this study was to quantitatively estimate the seasonal variation in the $f_{PV}$, $f_{NPV}$ and the fractional coverage of bare soil ($f_{BS}$) in the lower reaches of the Tarim River based on Sentinel-2A image data. The study is divided into the following sections: (1) select the best PVI and NPVI as estimation index of $f_{PV}$ and $f_{NPV}$ according to the correlation coefficient of the PVIs with $f_{PV}$ and NPVIs with $f_{NPV}$ by linear regression models analysis, (2) build pixel unmixing model based on Sentinel-2A image data and quantitatively evaluate the results of estimating $f_{PV}$ and $f_{NPV}$ by using field sampled data and (3) map different periods of $f_{PV}$, $f_{NPV}$ and $f_{BS}$ for the lower reaches of the Tarim River and analyze seasonal variation.

## 2. Materials and Methods

### 2.1. Study Area

The study area was located between the Dashkol Reservoir and Tetima Lake [39.5–40.59° N, 87.56–88.46° E] in the lower reaches of the Tarim River (Figure 1), which was surrounded by the Taklamakan Desert in the west and the Kuruktagh Desert in the northeast [29]. The study area's annual precipitation ranged from 20 to 50 mm but the annual potential evaporation varied from 2500 to 3000 mm [30,31]. The total annual solar radiation varied in a range of 5692–6360 MJ m$^{-2}$ with an annual sunshine from 2780 to 2980 h [32]. The water supply of the vegetation in the area mainly depended on the water streamed from the upper reaches of the river. The vegetation was mostly distributed on the floodplain on the riverbank, which composed of trees, shrubs and herbs. Dominant trees included the *Populus euphratica*, *Elaeagnus angustifolia*, the shrubs consisted of the *Tamarix ramosissima*, *Lycium ruthenicum*, *Halimodendron halodendron*, and herbs *Phragmites australis*, *Alhagi sparsifolia*, *Poacynum hendersonii*, *Karelinia caspia* [2].

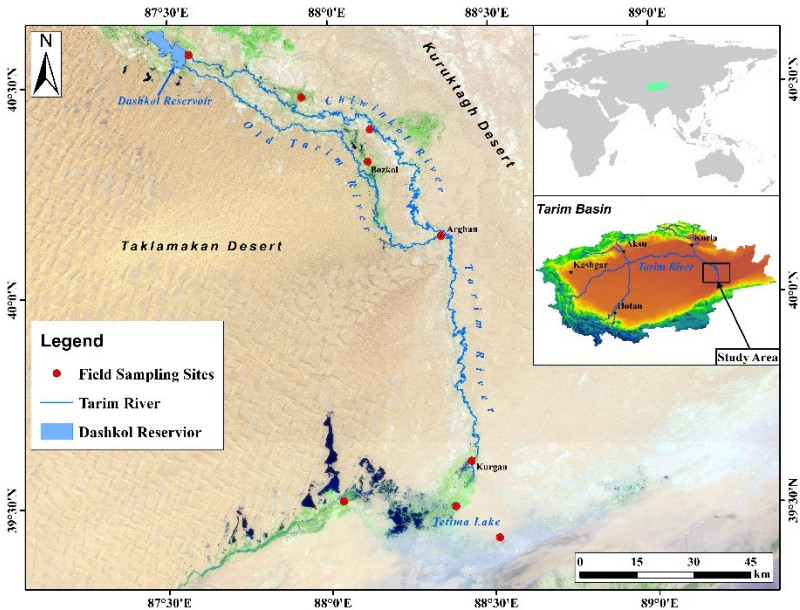

**Figure 1.** Location of the study area and distribution of the sample sites.

### 2.2. Datasets

#### 2.2.1. Field Data

The fractional coverage samples of the PV, NPV and bare soil (BS) were collected from the Dashkol Reservoir, Gancaochang, Bozkol, Arghan, Kurgan and Tetima Lake. All samples were collected from 25 September to 29 September 2020 when the PV, NPV and BS existed simultaneously. Data collection and processing steps were as follows: we defined 10 m × 10 m squares, aligned to the north, and covered by a homogeneous vegetation distribution. The four corners and the center of the square were precisely located with GPS. Secondly, a smaller square of 1 m × 1 m was positioned randomly 3–5 times in the

larger square. At the same time, a digital camera was used to take pictures vertically 1.6 m above the sample square. Each image was processed in ENVI 5.3. Photos were divided into NPV, PV and BS categories and training samples were established for each for supervised classification. Finally, we calculate the $f_{PV}$, $f_{NPV}$ and $f_{BS}$ of all 1 m × 1 m squares in each 10 m × 10 m square, and count the average value to represent the $f_{PV}$, $f_{NPV}$ and $f_{BS}$ in the 10 m × 10 m square. Figure 2 shows the field square acquisition and classification processing.

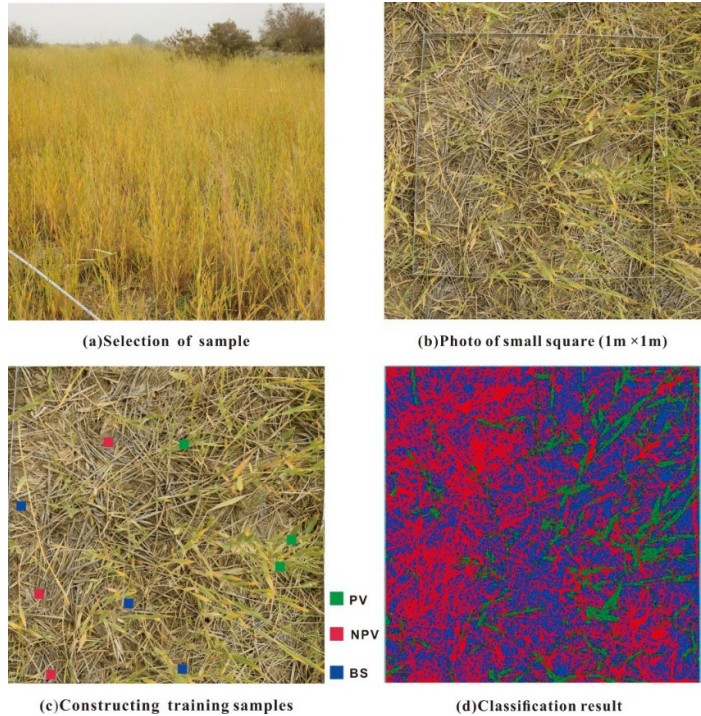

**Figure 2.** Field square acquisition and classification processing.

### 2.2.2. Remote Sensing Data

Sentinel-2A Level-1C data was used in this study. Images were from, seven periods on 5 April, 25 May, 24 July, 18 August, 12 September, 2 October and 6 November 2020. The data were radiometrically calibrated and geometrically corrected, downloaded from the ESA website (https://sci hub.copernicus.eu/home). The image preprocessing was carried out in SNAP 6.0 provided by ESA (Sentinel application platform) and the ENVI 5.3 software platform. The SNAP 6.0 Sen2Cor (Sentinel to Correction) plug-in was used to perform atmospheric correction processing on all L1C data, which resulted in 9 bands (Band 2, Band 3, Band 4, Band 5, Band 6, Band 7. Band 8a, Band 11 and Band 12) L2A surface reflectance data. And the 9-band data were combined into a single multiple image using composite bands in Quantum GIS (QGIS) and then resampled to 10 m pixels. Image data were mosaicked, and the study area was clipped. Water features were masked by using the modified normalized difference water index (MNDWI) combined with the threshold method (threshold value is −0.08).

### 2.3. Methods

The general workflow of this project was: (1) construct linear regression models of the PVIs and $f_{PV}$, NPVIs and $f_{NPV}$, respectively, to select the optimal PVIs and NPVIs, (2) analyze the feasibility of the selected PVIs and NPVIs so as to construct the response space, (3) apply the pixel linear unmixed model to determine the end member values of the PV, NPV and BS based on the Sentinel-2A image data, (4) quantitatively evaluate the estimation accuracy of the $f_{PV}$ and $f_{NPV}$ using the field sampled data, (5) map time series of

the $f_{PV}$, $f_{NPV}$ and $f_{BS}$ for the lower reaches of the Tarim River and to analyze the seasonal variation. The flowchart is shown in Figure 3.

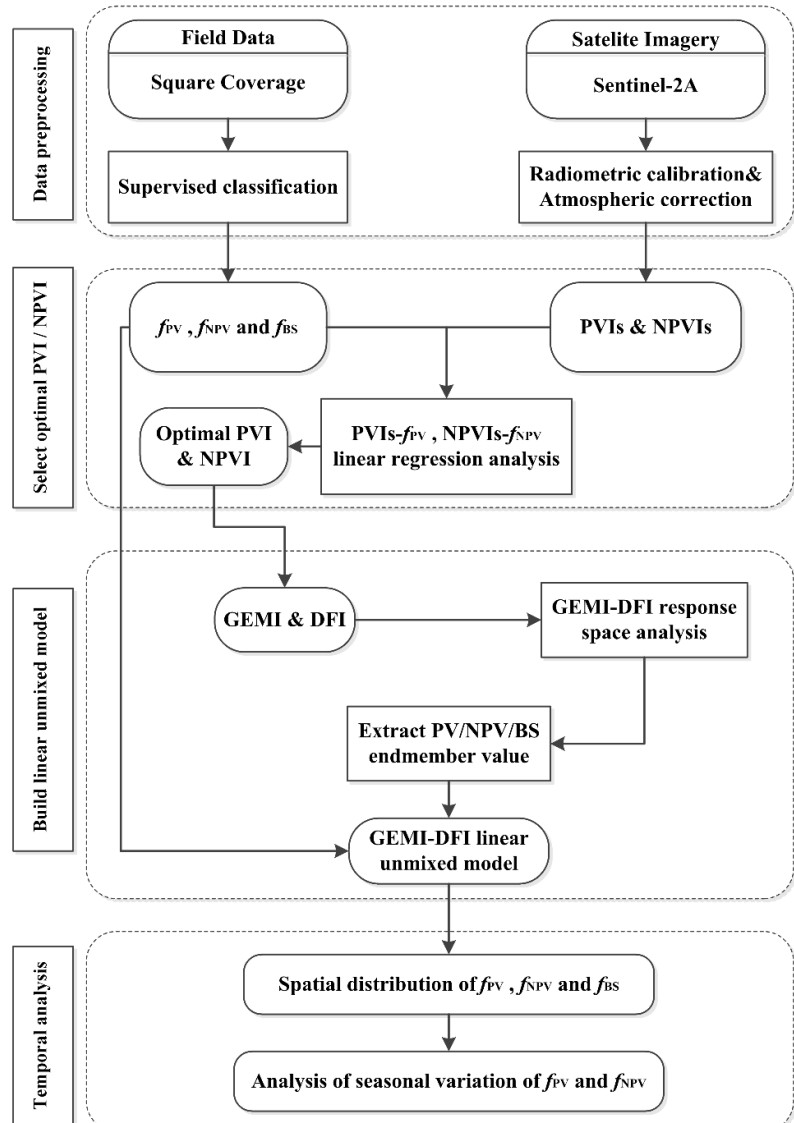

**Figure 3.** A methodology flowchart.

### 2.3.1. PVIs and NPVIs

PV was affected by chlorophyll and the cell structure and showed typical spectral characteristics of green vegetation, with obvious peaks and valleys [25]. In this study, we selected several PVIs for testing, including the NDVI [33], RVI [34], SAVI [35], MSAVI [16], RSR [36] and GEMI [17] (Table 1).

**Table 1.** The Sentinel-2-based multispectral photosynthetic vegetation indices (PVIs) used for the photosynthetic vegetation ($f_{PV}$) estimation.

| Vegetation Index | Equation | Citation |
|---|---|---|
| NDVI (normalized difference vegetation index) | $NDVI = \frac{R_{NIR} - R_{Red}}{R_{NIR} + R_{Red}}$ | Deering. 1978 |
| RVI (ratio vegetation index) | $RVI = \frac{R_{Red}}{R_{NIR}}$ | Jordan. 1969 |
| SAVI (soil adjusted vegetation index) | $SAVI = \frac{R_{NIR} - R_{Red}}{R_{NIR} + R_{Red} + L}(1 + L)$ | Huete. 1988 |
| MSAVI (modified soil adjusted vegetation index) | $MSAVI = \frac{2 \times R_{NIR} + 1 - \sqrt{(2 \times R_{NIR} + 1)^2 - 8 \times (R_{NIR} - R_{Red})}}{2}$ | Qi et al., 1994 |
| RSR (reduced simple ratio index) | $RSR = \frac{R_{NIR}}{R_{Red}} \times \left(1 - \frac{R_{SWIR1} - R_{SWIR1min}}{R_{SWIR1max} - R_{SWIR1min}}\right)$ | Brown et al., 2000 |
| GEMI (global environment monitoringindex) | $GEMI = \eta \times (1 - 0.25\eta) - \frac{R_{Red} - 0.125}{1 - R_{Red}};$ $\eta = 2 \times \frac{(R_{NIR}^2 - R_{Red}^2) + 1.5 \times R_{NIR} + 0.5 \times R_{Red}}{R_{NIR} + R_{Red} + 0.5}$ | Pinty et al., 1992 |

Notes: $R_{Red}$, $R_{NIR}$, $R_{SWIR1}$ and $R_{SWIR2}$ represent the reflectance of the Red band, Near Infrared band, Shortwave Infrared 1 band (1600 nm) and Shortwave Infrared 2 band (2100 nm), corresponding to the band 4, 8, 11 and 12 of sentinel-2A data, respectively. *L* is a soil adjustment factor, based on the experience *L* = 0.5.

NPV and BS had similar spectral reflectance characteristics in Visible and Near Infrared band and but could be differentiated using the Shortwave Infrared band (1600 nm and 2100 nm) [25]. In this study, we selected several PVIs for testing, including the NDI [23], NDTI [19], NDSAVI [20], STI [19], SWIR32 [3] and DFI [24] (Table 2).

**Table 2.** The Sentinel-2-based multispectral non-photosynthetic vegetation indices (NPVIs) used for the non-photosynthetic vegetation ($f_{NPV}$) estimation.

| Vegetation Index | Equation | Citation |
|---|---|---|
| NDI (normalized difference index) | $NDI = \frac{R_{NIR} - R_{SWIR1}}{R_{NIR} + R_{SWIR1}}$ | Mc Nairn et al., 1993 |
| NDTI (normalized difference tillage index) | $NDTI = \frac{R_{SWIR1} - R_{SWIR2}}{R_{SWIR1} + R_{SWIR2}}$ | Deventer et al., 1997 |
| NDSVI (normalized difference senescent vegetation index) | $NDSVI = \frac{R_{SWIR1} - R_{Red}}{R_{SWIR1} + R_{Red}}$ | Qi et al., 2002 |
| STI (soil tillage index) | $STI = \frac{R_{SWIR1}}{R_{SWIR2}}$ | Deventer et al., 1997 |
| SWIR32 (shortwave infrared ratio) | $SWIR32 = \frac{R_{SWIR2}}{R_{SWIR1}}$ | Guerschman et al., 2009 |
| DFI (dead fuel index) | $DFI = 100 \times \left(1 - \frac{R_{SWIR2}}{R_{SWIR1}}\right) \times \frac{R_{Red}}{R_{NIR}}$ | Cao et al., 2010 |

Notes: $R_{Red}$, $R_{NIR}$, $R_{SWIR1}$ and $R_{SWIR2}$ demonstrate the reflectance of the Red band, Near Infrared band, Shortwave Infrared 1 band(1600 nm) and Shortwave Infrared 2 band (2100 nm), corresponding to the band 4, 8, 11 and 12 of sentinel-2A data, respectively.

PVIs and NPVIs were calculated using ArcGIS 10.5 software. The PVI value and $f_{PV}$ measured value, and NPVI value and $f_{NPV}$ measured value, were constructed linear regression models of PVI and NPVI relative to $f_{PV}$ and $f_{NPV}$, respectively, and were used to evaluate the accuracy of the various indices. Considering the number of samples we have taken, we used the leave-one-out cross-validation (LOOCV) which was suitable for a small number of sample. In LOOCV, each sample was excluded in turn and the regression model was calculated with all the remnants samples and used to predict that sample. The benefit of LOOCV was its aptitude to detect outliers and its capability of providing nearly unbiased estimations of the prediction error [4,37]. The performance of these models was assessed by the coefficients of determination ($R^2$), root mean square error of leave-one-out cross-validation (*RMSECV*) and regression significance (*p*):

$$R^2 = \frac{\sum_{i=1}^{n}(x_i - \overline{x})(y_i - \overline{y})}{\sum_{i=1}^{n}(x_i - \overline{x})^2 \sum_{i=1}^{n}(y_i - \overline{y})^2} \tag{1}$$

$$RMSECV = \sqrt{\frac{1}{n} \sum_{i=1}^{n} (x_i - y_i)^2} \qquad (2)$$

where $n$ shows the sample plots' number, $x_i$ is the measured value of the sample plot $i$, $y_i$ stands for the estimated value of the sample plot $i$, $\overline{x}$ illustrates the mean value of the measured sample plots, $\overline{y}$ stipulates the mean value of the estimated sample plots.

### 2.3.2. Linear Unmixed Model

Guerschman [3] hypothesized that the NDVI and CAI could resolve the fractions of the PV, NPV and BS, when the NDVI and $f_{PV}$ were linearly related (as were the CAI and $f_{NPV}$). This situation is reflected in the scatterplot of the NDVI and CAI, namely that the feature space forms a triangle; BS is situated on the right side of the triangle, with a high NDVI and intermediate CAI value; the NPV is located in the upper left corner of the triangle, showing a low NDVI and a high CAI value; the BS is seen in the lower left corner of the triangle, having low NDVI and CAI values; Cao [24] used the DFI (replacing CAI) to present the NPV in order to construct the NDVI-DFI linear unmixed model, and it is successful in estimating the fractions of the PV, NPV and BS. Therefore, we applied the GEMI-DFI linear unmixed model to assess the fractions of the PV, NPV and BS in the study area (Figure 4).

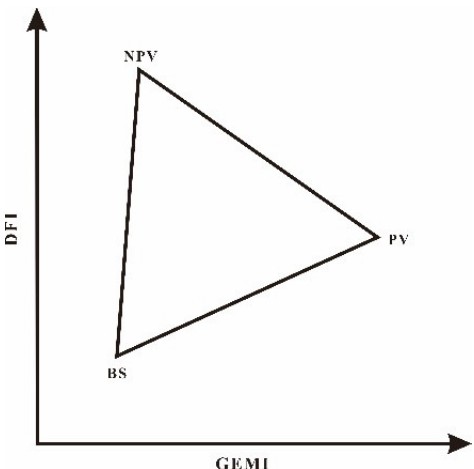

**Figure 4.** The conceptual model of the global environment monitoring index and cellulose absorption index (GEMI-CAI) relation triangle of the PV, NPV and BS.

PV is expressed by the global environmental monitoring index (GEMI) and the NPV by the dead fuel index (DFI), The GEMI formula is calculated as Table 1, The DFI formula is the following in Table 2. The relative proportions of each fractional coverage for any Sentinel-2A image pixel were found by solving the equations:

$$G_S = \sum[f_i G_i] = [f_{PV} G_{PV} + f_{NPV} G_{NPV} + f_{BS} G_{BS}] \qquad (3)$$

$$D_S = \sum[f_i D_i] = [f_{PV} D_{PV} + f_{NPV} D_{NPV} + f_{BS} D_{BS}] \qquad (4)$$

$$\sum f_i = [f_{PV} + f_{NPV} + f_{BS}] = 1 \qquad (5)$$

where $G_S$ and $D_S$ show the GEMI and DFI value in the given Sentinel-2A image pixel, the $f_{PV}, f_{NPV}$ and $f_{BS}$ illustrate the fractional coverage of the PV, NPV and BS; the $G_{PV}, G_{NPV}$ and $G_{BS}$ are the GEMI values of the end members, the $D_{PV}, D_{NPV}$ and $D_{BS}$ demonstrate the DFI values of the end members. It forces the values of $f_{PV}, f_{NPV}$ and $f_{BS}$ to sum to unity. If the sum does not get one, the pixel has a negative or higher than 1 value in at least one end member. When that occurred, the following correction was applied:

$$C_x = 0(-0.2 < C_x < 0) \qquad (6)$$

$$C_x = 1 (1 < C_x < 1.2) \tag{7}$$

$$C_y = C_y / (C_y + C_z) \tag{8}$$

$$C_z = C_z / (C_y + C_z) \tag{9}$$

where $C_x$ is the value not within a specified range ($C_x < -0.2$ or $C_x > 1.2$), and $C_y$ and $C_z$ are the values of the other two end members. If the range of $C_x$ is amounts from $-0.2$ to $0$, $C_x$ is 0; if the range of $C_x$ is 1 to 1.2, $C_x$ is 1; the condition is mentioned in the above two cases, we only calculated $C_y$ and $C_z$.

### 2.3.3. Determination of the GEMI and DFI End Member Value

The choice of pure end members was the key to success of the PV and NPV model inversion. The GEMI-DFI linear unmixed model needed to become a pure end member of the PV, NPV and BS to calculate the corresponding proper values and the model needed be used to solve the fractional coverage of each end member. We employed the pixel purity index (PPI) method to determine the pure end member. Firstly, the Sentinel-2A images were subjected to the minimum noise fraction (MNF) so as to reduce the image dimensionality during different periods. The first 6 bands of each image period were selected for calculation and the number of iterations was set to 5000 in order to generate the PPI. Secondly, the GEMI and DFI were calculated for various period images. Finally, the pixels near the vertices of the triangular feature space were regarded as pure pixels.

### 2.3.4. Model Evaluation

In order to quantitatively evaluate the performance of the model's estimation ability, the coefficients of determination ($R^2$), root mean square error (*RMSE*) and mean error (*ME*) were applied.

$$RMSE = \sqrt{\frac{1}{n} \sum_{i=1}^{n} (x_i - y_i)^2} \tag{10}$$

$$ME = \frac{1}{n} \sum_{i=1}^{n} (x_i - y_i) \tag{11}$$

where $n$ shows the sample plots' number, $x_i$ is the measured value of the sample plot $i$, $y_i$ stands for the estimated value of the sample plot $i$.

## 3. Results

### 3.1. PVI and NPVI Index Optimization

In this study, six PV indices (NDVI, RVI, SAVI, MSAVI, RSR and GEMI) were selected to build linear regression models with $f_{PV}$ (Table 3 and Figure 5). Various PVIs exhibited different performance, with the $R^2$ ranging from 0.33 to 0.59 and the *RMSECV* ranging from 0.752 to 0.1283. The GEMI index showed the highest correlation, with the $R^2$ reaching 0.59 and *RMSECV* being 0.752 ($p < 0.05$). The RVI, MSAVI and RSR indices had low correlations (all $R^2$ lower than 0.50 and *RMSECV* higher than 0.8). The GEMI index was used as the PVI index to estimate the $f_{PV}$ of the study area. In order to count the GEMI value, the GEMI index value had been normalized.

**Table 3.** Parameters of linear regression analyses between PVIs and $f_{PV}$.

| PVIs | Regression Equation | $R^2$ | *RMSECV* | $p$ |
|------|---------------------|-------|----------|-----|
| NDVI | $y = 1.4562x + 0.1199$ | 0.51 | 0.0815 | $p < 0.05$ |
| RVI | $y = 0.9331x + 0.7924$ | 0.47 | 0.0817 | $p < 0.05$ |
| SAVI | $y = 1.1202x - 0.1199$ | 0.51 | 0.0795 | $p < 0.05$ |
| MSAVI | $y = 0.9333x - 0.1407$ | 0.47 | 0.0814 | $p < 0.05$ |
| RSR | $y = 0.4316x - 0.2178$ | 0.33 | 0.1283 | $p < 0.05$ |
| GEMI | $y = 2.8495x - 0.0349$ | 0.59 | 0.0752 | $p < 0.05$ |

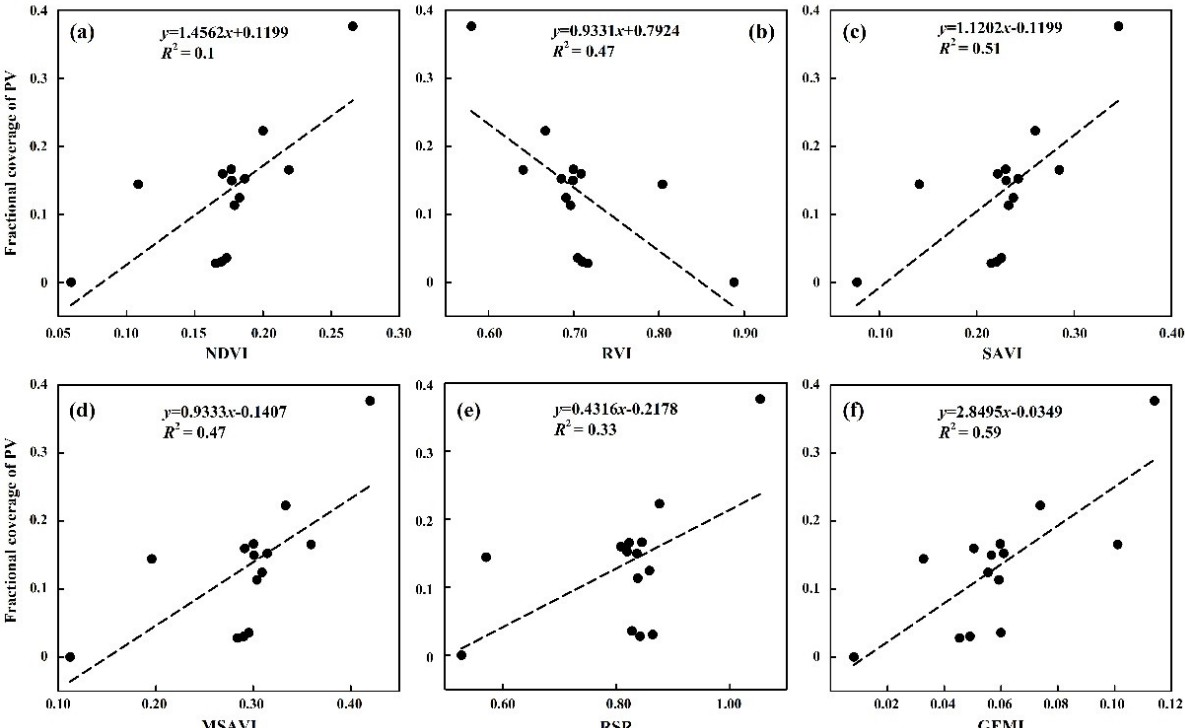

**Figure 5.** Linear regression between different PVIs and the $f_{PV}$. The PVIs are (**a**) NDVI, (**b**) RVI, (**c**) SAVI, (**d**) MSAVI, (**e**) RSR and (**f**) GEMI, respectively.

We selected six NPVI indices (NDI, NDTI, NDSAVI, DFI, STI and SWIR32) to construct linear regression models with the $f_{NPV}$ (Table 4 and Figure 6). Compared with the correlation between the PVI indices and $f_{PV}$, the correlation between the NPVIs and $f_{NPV}$ was lower. This might be caused by similar spectral characteristics of the NPV and BS [24]. The DFI index showed the best performance with a $R^2$ of 0.45 and *RMSECV* of 0.2111($p < 0.05$). The STI, NDTI, SWIR32, NDSVI and NDI indices were lower than the DFI, with the $R^2$ ranging from 0.02 to 0.43, and the *RMSECV* from 0.2611 to 0.2627. Therefore, the DFI index was selected to assess the $f_{NPV}$ of the study area.

**Table 4.** Parameters of linear regression analyses between NPVIs and $f_{NPV}$.

| NPVIs | Regression Equation | $R^2$ | *RMSECV* | *p* |
|---|---|---|---|---|
| NDI | $y = 0.9950x + 0.4855$ | 0.02 | 0.2627 | $p < 0.05$ |
| NDTI | $y = 3.4457x + 0.1076$ | 0.39 | 0.2223 | $p < 0.05$ |
| NDSVI | $y = 3.1160x + 0.9570$ | 0.11 | 0.2606 | $p < 0.05$ |
| DFI | $y = 0.0328x - 0.0422$ | 0.45 | 0.2111 | $p < 0.05$ |
| STI | $y = 1.4480x - 1.3296$ | 0.43 | 0.2161 | $p < 0.05$ |
| SWIR32 | $y = -2.0018x + 2.0096$ | 0.37 | 0.2272 | $p < 0.05$ |

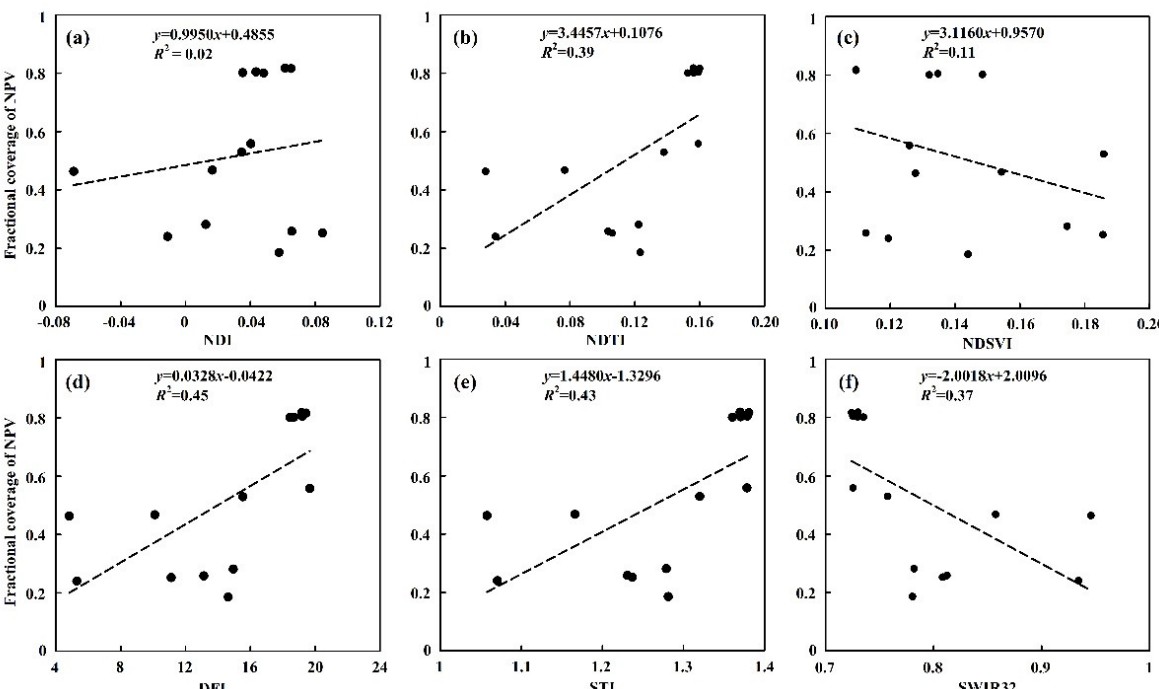

**Figure 6.** Linear regression between different NPVIs and the $f_{NPV}$. The NPVIs are (**a**) NDI, (**b**) NDTI, (**c**) NDSVI, (**d**) DFI, (**e**) STI and (**f**) SWIR32, respectively.

### 3.2. The Feasibility of the GEMI-DFI Model

We obtained the GEMI-DFI response space by calculating the GEMI and DFI values for seven Sentinel-2A images' data (Figure 7). GEMI-DFI response space of the first six images was basically triangular, which was consistent with the theoretical conceptual mode. It showed that the GEMI-DFI linear unmixed model could be utilized to estimate the fractional coverage of the PV, NPV and BS. The last GEMI-DFI response space image did not seem to conform to the triangle shape. We selected images covering the growing season from spring to fall (April to November) and as vegetation leafed out and became active, GEMI values increased while DFI values declined. The inverse was seen with the onset of fall, as vegetation began to senesce and go dormant.

### 3.3. Evaluation of the $f_{PV}$ and $f_{NPV}$ Estimation Accuracy

Considering that field data collection occurred between 25–29 September in 2020, Sentinel 2A data collected on 2 October in 2020 were selected. We used the GEMI-DFI linear unmixed model (combined with the PPI) to obtain the spatial distribution of the $f_{PV}$, $f_{NPV}$ and $f_{BS}$. The BS proportion in the study area was the largest and the PV and NPV were mainly distributed along the downstream river and the Tetima Lake. In order to clearly show the model result in estimating the $f_{PV}$ and $f_{NPV}$, we selected the areas with a vegetation distribution evenly for display. The four regions were: a (Dashkol), b (Chiwinkol), c (Bozkol) and d (Tetima Lake). The $f_{NPV}$ of the four regions was greater than the $f_{PV}$, indicating that after October, the vegetation entered the end of the growing season and the increase in the NPV (during this period) caused an increase in the proportion of the $f_{NPV}$ (Figure 8) in turn.

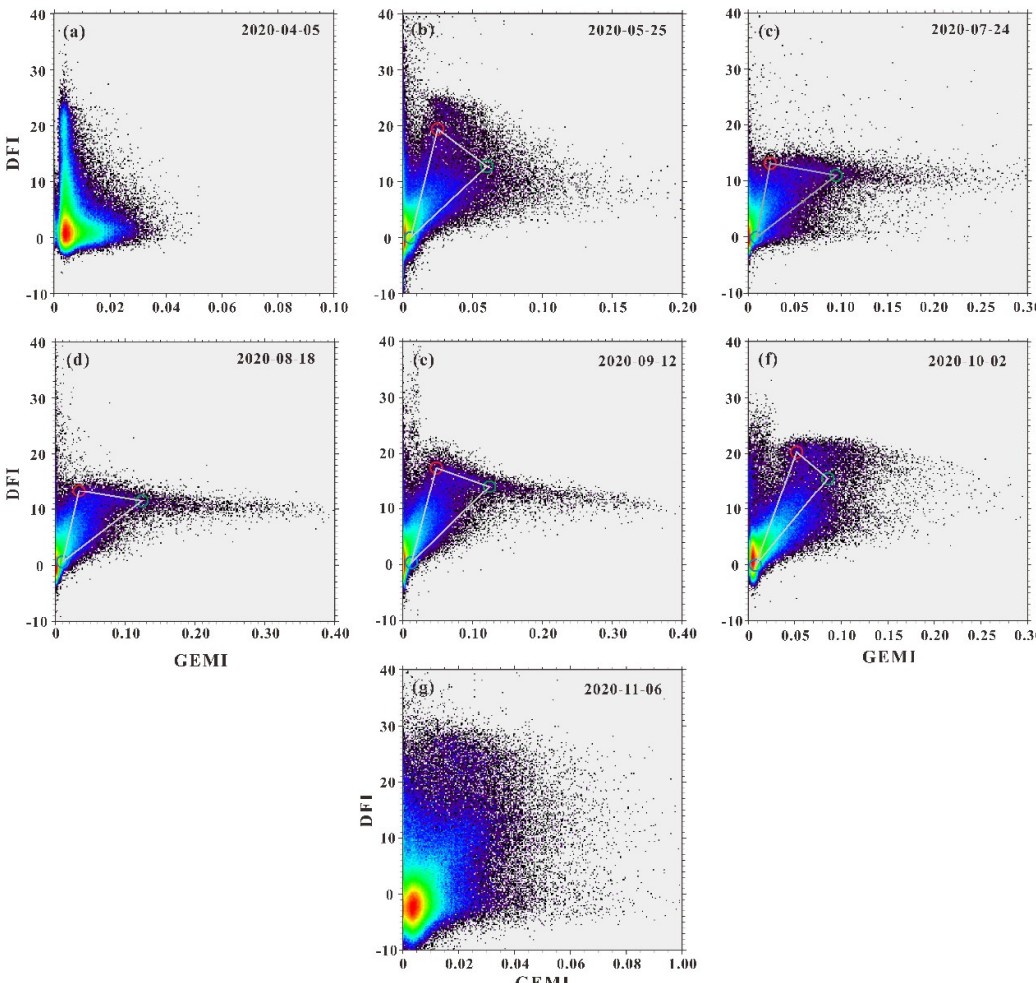

**Figure 7.** Response spaces of the Sentinel-2A reflectance spectral defined by the GEMI and DFI for seven dates in 2020. The green, red and blue circle indicate the position of the PV, NPV and BS end members, respectively (and form triangles). The colors indicate the point density from blue (low density) to dark red (high density). (**a**) 5 April 2020, (**b**) 25 May 2020, (**c**) 24 July 2020, (**d**) 18 August 2020, (**e**) 12 September 2020, (**f**) 2 October 2020, (**g**) 6 November 2020.

We could conclude from Figure 9 that the estimated $f_{PV}$ has the best correlation with the measured $f_{PV}$, with a $R^2$ of 0.69 and *RMSE* of 0.07 ($p < 0.05$). The estimated $f_{NPV}$ and the measured $f_{NPV}$ have a lower correlation with a $R^2$ of 0.58 and *RMSE* of 0.17 ($p < 0.05$). For BS, the measured (and estimated) values possess the lowest correlation, with a $R^2$ of 0.43 and a *RMSE* of 0.17 ($p < 0.05$). For the PV, the fitted line is situated below the reference line (1:1 line) and the *ME* value amounts to $-2.67\%$. The data mentioned above show that overall the estimated $f_{PV}$ value is lower than the measured $f_{PV}$ value, this is due to the fact that the acquisition time of the Sentinel-2A image data is later than the one for the field measured data, resulting in lower $f_{PV}$ estimates. Concerning the $f_{NPV}$, the value of *ME* is 3.14%, this means that the estimated $f_{PV}$ value is higher than the measured $f_{PV}$ value. In general, the estimation accuracy of the $f_{NPV}$ and $f_{BS}$ is lower than the $f_{PV}$, this reflects the fact that the PV could be resolved in an easier way than the NPV. Both the NPV and BS have similar spectral reflectance characteristics, resulting in a greater probability of wrong classification between the NPV and BS.

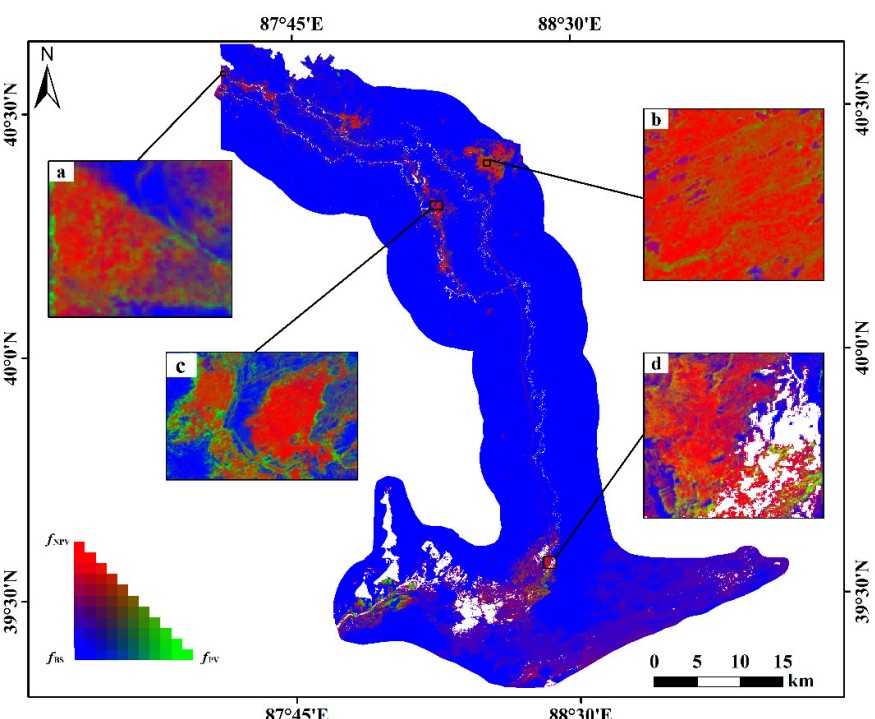

**Figure 8.** Spatial distribution of the $f_{PV}$, $f_{NPV}$ and $f_{BS}$ in the study area (based on the Sentinel-2A image). Four areas: a (Dashkol), b (Chiwinkol), c (Bozkol) and d (Tetima Lake). The proportions of the fractional coverage are shown in the RGB. The white area of the map represents the location in which the presence of a water body has been masked.

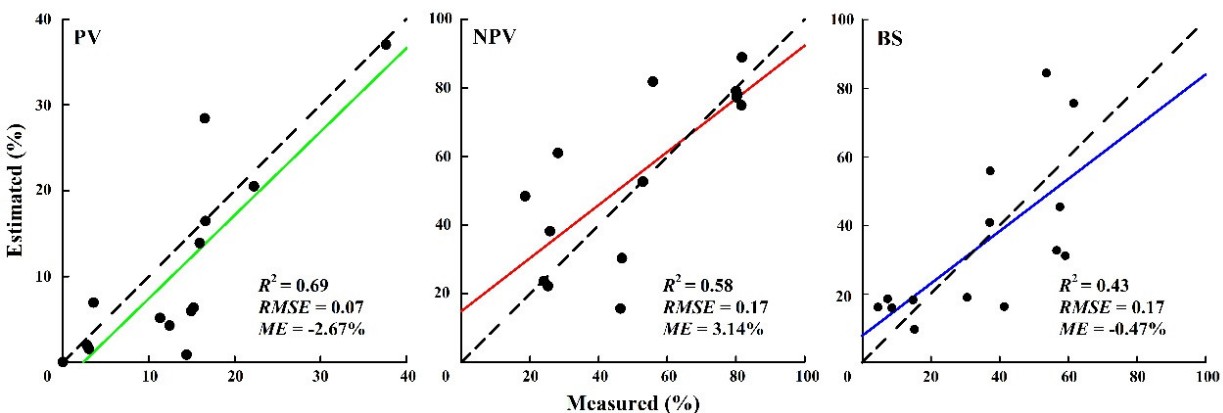

**Figure 9.** Estimating the accuracy of the $f_{PV}$, $f_{NPV}$ and $f_{BS}$ from the field measurements. The dotted line represents the 1:1 reference line. The green, red and blue solid lines denote the fitted lines of the PV, NPV and BS, respectively.

### 3.4. Seasonal Variation of the $f_{PV}$ and $f_{NPV}$

In this study, the b region was chosen as the representative area for seasonal variation. This was located in the Chiwinkol wetland, which was less disturbed by human activities and could have better indicated the natural alternation of the PV and NPV's seasonal variation. We selected Sentinel-2A image data from seven periods and used the GEMI-DFI linear unmixed model to estimate the $f_{PV}$, $f_{NPV}$ and $f_{BS}$ during different periods (Figure 10). We counted mean values of the $f_{PV}$, $f_{NPV}$ and $f_{BS}$ in the *b* region (Figure 11). By analyzing the changes of the $f_{PV}$, $f_{NPV}$ and $f_{BS}$ values, it was found that the seasonal variation of the $f_{PV}$ and $f_{NPV}$ was estimated by the GEMI-DFI model in conformity with the characteristics of the vegetation phenology. In April, most of the vegetation not yet broken dormancy and

as such, there was a large NPV amount, the GEMI value was low, the DFI value was high, the $f_{NPV}$ was 0.93 and the $f_{PV}$ measured 0.01. In May, as the vegetation had emerged and begun actively growing, the GEMI value increased, the DFI value decreased, the $f_{PV}$ rose to 0.17 and the $f_{NPV}$ declined to 0.69. As the growing season progressed, GEMI increased, peaking in August with DFI following an inverse trend, and the DFI value fell to the lowest point. Similarly, $f_{PV}$ reached maximum value, which was 0.44 and $f_{NPV}$ its lowest value of 0.45. In September, the vegetation began to yellow and GEMI fell and DFI rose. By the beginning of October, most of the vegetation had already withered, the $f_{NPV}$ occupied the dominant position again (at 0.81) and the $f_{PV}$ decreased further to 0.17. The $f_{BS}$ value remained in a relatively stable position throughout the whole year.

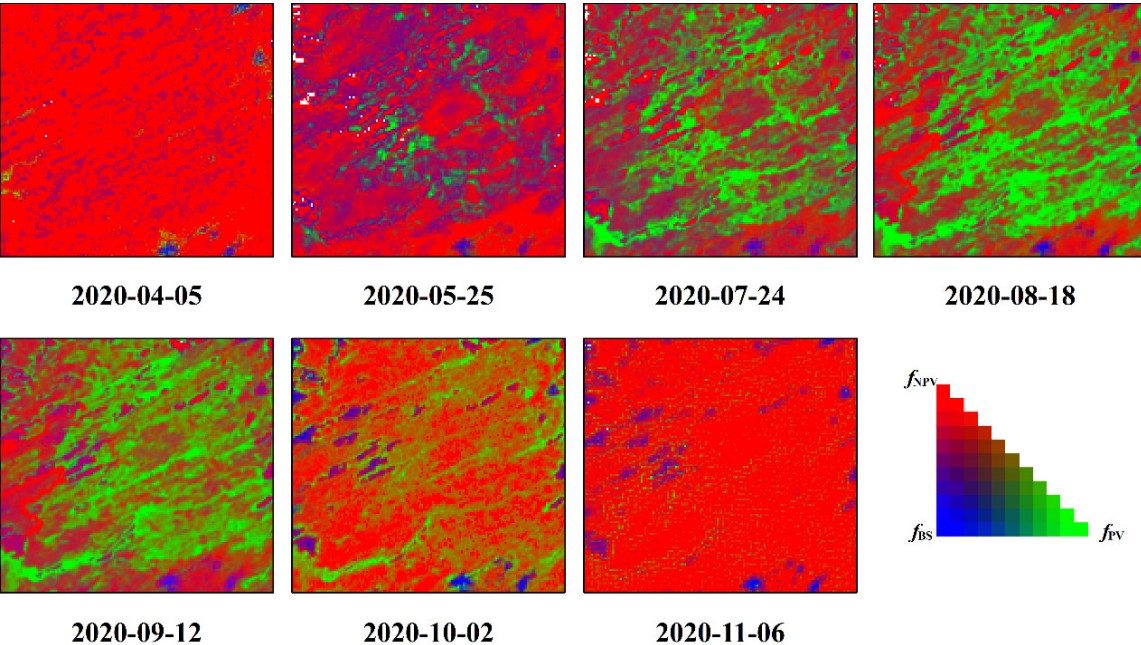

**Figure 10.** Spatial distribution of the $f_{PV}$, $f_{NPV}$ and $f_{BS}$ in the b region in 2020. The proportions of the fractional coverage are shown in the RGB. The white map area is the location in which the presence of a water body has been masked.

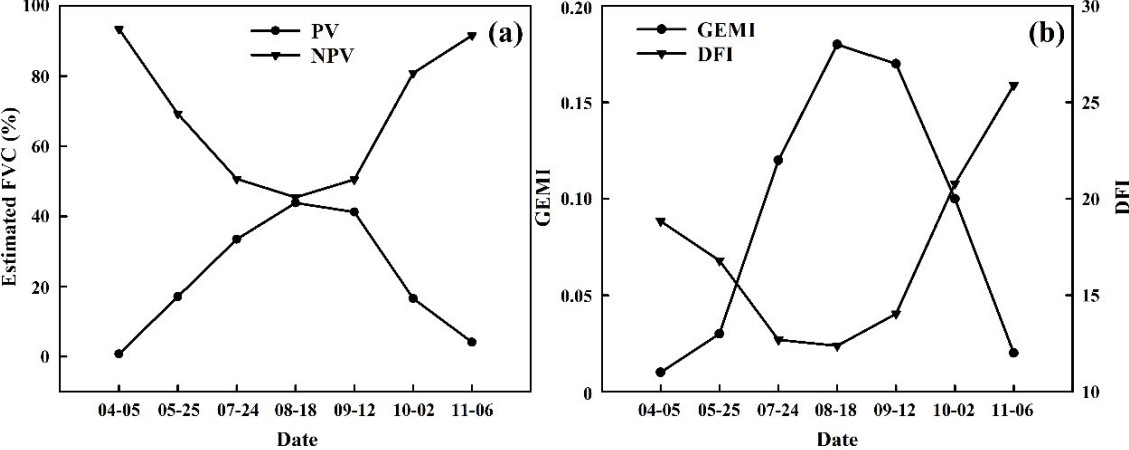

**Figure 11.** (**a**) Temporal variation of the $f_{PV}$ and $f_{NPV}$ in the area of b. (**b**) Temporal variation of the GEMI and DFI values in the *b* region.

## 4. Discussion

This study explored the performance of several Sentinel-2A based PVIs by constructing linear regression models in the lower reaches of the Tarim River. The GEMI index has the best correlation with the $f_{PV}$, which is consistent with the conclusion of Liu [38] on the vegetation information monitoring this area. The GEMI index shows greater advantages in detecting the low coverage vegetation information. In the regression analysis between the NPVI index and $f_{NPV}$, the correlation between the DFI index and $f_{NPV}$ is better than for those of the NDI, NDTI, NDSVI, STI and SWIR32. Among them, the NDI and NDSVI indices mainly use Band 11 (SWIR1: 1565~1655 nm), combined with a visible light band (NDI for the NIR band, NDSVI for the Red band) and without combining Band 12 (SWIR: 2100~2280 nm). Previous research has demonstrated that the cellulose absorption signature (amounting to around 2100 nm in the SWIR region) has a strong discriminatory ability between the NPV, PV and BS [8]; so Band 12 is a good choice to distinguish the PV, NPV and BS. Compared with the DFI index, STI and SWIR32 simply perform ratio calculations. The DFI index uses more bands for the calculation, including the Red, NIR SWIR1 and SWIR2. It might reduce the impact of the BS background to some extent and it might further improve NPV detection [4]. Ji [39] similarly concluded that the red-edge and NIR bands of the Sentinel-2 data are effective in improving the accuracy of the $f_{NPV}$ estimates. The manner in which the red-edge band of the Sentinel-2 data could be utilized in order to estimate the vegetation information accurately still need further exploration.

The study uses the PPI method to extract the pure end members of the image data, which eliminates the discrepancies between the image data and the measured data, ensuring that both have the same spatial scale, which is widely employed to select the pure end of the spectrum [14,37]. We were able to estimate the $f_{PV}$ and $f_{NPV}$ of the study area in different periods by means of the GEMI-DFI model, revealing the changes in the photosynthetic and non-photosynthetic vegetation at different growth stages. The NPV proportion is greater at the beginning and at the end of the growing season, while the proportion of the PV peaks in the middle of the vegetation growing season. This is consistent with the findings of Wang [40] estimating the $f_{PV}$ and $f_{NPV}$ in the typical grasslands of Xilingol, indicating that the GEMI-DFI model is feasible to estimate the $f_{PV}$ and $f_{NPV}$ in the lower reaches of the Tarim River. The accuracy of the $f_{NPV}$ estimation ($R^2 = 0.58$) is slightly lower than for the $f_{PV}$ estimation ($R^2 = 0.69$). As the NPV and BS have similar spectral reflection characteristics in the visible band [3,41], the NPV estimation is more susceptible to the influence of the BS background, which could lead to a false distinction between the NPV and BS. Both the GEMI and DFI values are influenced by a variety of factors, such as the vegetation type, vegetation structure, etc. [17,24]. Therefore, more consideration should be given to the influencing factors in future studies on the PV and NPV.

The overall vegetation coverage in the study area is low, as vegetation is mostly distributed along the watercourses. Due to this, higher spatial resolution imagery is required to obtain more detailed information on the type of the vegetation [42]. Based on this, the Sentinel-2A data with a resolution of 10 m were used, which might reduce the spatial heterogeneity of the mixed pixels and improve the estimation accuracy by a better acquisition of pure pixels. There is still some uncertainty noticeable in the estimation of the PV, NPV and BS using multispectral data. In fact, the vegetation and other surrounding components form a complex system [37] and it endures stages of greening, heading, flowering, maturing and yellowing [43]. The above-mentioned conditions affect the acquisition of the estimated GEMI and DFI values, thereby impacting the model's accuracy to estimate the $f_{PV}$, $f_{NPV}$ and $f_{BS}$. In addition, uncertain factors also exist in the field data collection. The moment in which the photo was taken does not match the time of the satellite transit; the visual interpretation of the PV, NPV and BS classification and the accuracy of the geometric alignment will inevitably be influenced by subjective factors, resulting in errors in the $f_{PV}$, $f_{NPV}$ and $f_{BS}$ estimates [18]. This study relies on the coverage obtained from field sampled data as the verification data. Taking into account the diversity of vegetation types and structures, the spectral characteristics of vegetation are different. We should combine

spectral data to accurately estimate the changing trends of different types of vegetation. The use of drones is also an effective measure to improve the estimation of vegetation coverage. This method can obtain a large area of vegetation coverage and reduce the error of its estimation and should be taken into account in future research work.

## 5. Conclusions

In this study, we constructed the GEMI-DFI linear unmixed model based on the Sentinel-2A image data so as to estimate the fractional coverage of the PV, NPV and BS in the lower reaches of the Tarim River combined with the field measured data for an accuracy evaluation. The main conclusions are as follows:

We established a linear regression model for the PVIs and $f_{PV}$, and NPVIs and $f_{NPV}$. The study found that the GEMI have a significantly linear correlation with the $f_{PV}$, $R^2$ is 0.59 and the DFI has a significantly linear correlation with the $f_{NPV}$, $R^2$ measuring 0.45. We used GEMI and DFI indices to construct linear unmixed model, the response space shown as triangle, which conformed the basic assumption of the linear unmixed model.

The GEMI-DFI linear unmixed models could effectively estimate the $f_{PV}$ and $f_{NPV}$, but the accuracy of estimation $f_{PV}$ is higher than that of $f_{NPV}$. How to improve the accuracy of estimation of $f_{NPV}$ is the focus of future work.

Considering the number of sampling data, we need to collect more vegetation coverage data in future work. We can use drones as a platform for obtaining vegetation coverage to achieve a wide range of coverage, thereby improving the accuracy of $f_{PV}$, $f_{NPV}$ and $f_{BS}$ estimation.

**Author Contributions:** Conceptualization, A.K.; methodology, Z.G.; software, A.K.; validation, Z.G., A.A. and A.K.; resources, A.K.; data curation, Z.G. and S.W.; writing—original draft preparation, Z.G.; writing—review and editing, Z.G., A.K., T.V.d.V., H.A., P.D.M. and E.D.U.; visualization, Z.G.; supervision, A.K.; project administration, A.K.; funding acquisition, A.K. All authors have read and agreed to the published version of the manuscript.

**Funding:** This research was funded by the National Natural Science Foundation of China (grant number 32071655; 31570536). Chinese Academy of Sciences President's International Fellowship Initiative (PIFI, Grant No. 2021VCA0004, 2017VCA0002).

**Acknowledgments:** The authors would like to thank ESA (https://scihub.copernicus.eu/ (accessed on 2 April 2021)) for providing Sentinel-2A data for this study. Thanks to Management Bureau of the Tarim River Main Stream for providing support for field work, especially Shengwu Xu and Salamu Ayoufu. Thanks to all the editors and reviewers for their meaningful comments so that we can improve the quality of this manuscript in different aspects.

**Conflicts of Interest:** The authors declare no conflict of interest.

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
