# Peer review of "Estimation of Photosynthetic and Non-Photosynthetic Vegetation Coverage in the Lower Reaches of Tarim River Based on Sentinel-2A Data"

_remotesensing, doi:10.3390/rs13081458_

Round 1

Reviewer 2 Report

-  in general, the text is good, expept some parts that should be improved in terms of stylistics.

- some useless repetitions: ex. formulas for GEMI and DFI are included twice (Table2 and 3 + rows 212-215)

- rows 222-233 are not clear in what they wanted to inform about. There is no letter/variable "C" mentioned within the text before and is not clear for what these calculations should serve for.

- some expressions do not sound very scientific, such as "We devided the photos into NPV, PV and BS through visual interpretation, and established a certain number of training samples..."

- some of the references are not cited correctly. Ex.: Table2 "Ji et al., 1994" does not exist in the list of references. It should be paired probably with "Qi et al. 1994"? Similarly, Verstraete et al., 1992 in Table 2 is probably Pinty, et al., 1992  in reality (?).

- there are many other “small” things to improve. Ex.: Figure 2 with the title: “The conceptual model of the NDVI-CAI relation triangle of the PV, NPV and BS.” but axis-labels in the picture itself shows GEMO (x-axis) and DFI (y-axis). I guess it remained as taken from other article, probably that of Guerschman [3].

- in Figure 3, regression models for NDVI and MSAVI (and SAVI) look very similar . Is it surely well calculated? I would expected that regression equations to be given in the article.

- The most important objection I have is the number of observations/areas that were examined for this purpose (and the quality level of this journal). 14 measurements are too low for this type of study (and this journal) and must be supplemented by others, so that you have at least 30 but better up to 50 observations and the results are scientifically reliable. Might be, that that case, you would chose different model as the best one to determine (non-)photosynthetic vegetation.

Reviewer 3 Report

1 As this is a algorithm validation paper. One or two paragraphs on the summary of current algorithms to calculate the pv, npv, and bs from satellite platforms are necessary in the introduction. These contents should be a detailed narration based on the lists of Table 2 and 3, but with performance comparison from the authors' point of view. 

2 With regards to all statistical modeling, my major concern is the limited sample number of in situ observation. I suggest you running normality test to make sure all models you adopted follow basic statistical assumptions in your case.

3 As your sample size is small, I suggest you trying non-parametric statistical  analysis (Tao or rho) to make sure the correlation between in situ and satellite data. These analysis could neglect the necessary parametric statistical assumption mentioned precedingly. 

4 In addition, you can try to kridge (with edge) based on these 14 observation in your whole target area, then compare the kridging results with satellite observations. 

Reviewer 4 Report

The paper needs some minor corrections according to errors in writing.
The Introduction and Discussion paragraphs also needs some more attention. 

The article should contain more references to other publications dealing with similar topics, e.g.:

Gurdak R., Bartold M., 2020, Remote sensing techniques to assess chlorophyll fluorescence in support of crop monitoring in Poland, Miscellanea Geographica - Regional Studies on Development, vol. 25, no. 3, ISSN: 2084-6118, DOI: 10.2478/mgrsd-2020-0029

The text should be refined.

Figure 1. -> a misspelling lenged -> legend

Reviewer 5 Report

The work presented here is sound and the logic behind it's interpretation is solid. The methodology and the analysis both look well planned out and executed. The only issue I had with the piece was the writing. In many places it was difficult to follow on first reading and the way some information was presented was confusing. I have included comments/suggestions should they been helpful. 

Author Response

Thank you very much for your revision of the language of my manuscript. I think it is very important for the literal expression of my manuscript. I have modified it according to your comments. I have uploaded my marked manuscript .Please see the attachment.

Thank you again for your great help in my manuscript. Thank you!

Round 2

Reviewer 1 Report

I have reread the last version ; it has been improved dramatically.

There are a few minor problems. Please check the format of reference.